# Sport Intervention Programs (SIPs) to Improve Health and Social Inclusion in People with Intellectual Disabilities: A Systematic Review

**DOI:** 10.3390/jfmk4030057

**Published:** 2019-08-15

**Authors:** Lidia Scifo, Carla Chicau Borrego, Diogo Monteiro, Doris Matosic, Kaltrina Feka, Antonino Bianco, Marianna Alesi

**Affiliations:** 1Department of Psychology, Educational Science and Human Movement, University of Palermo, IT, V.le delle Scienze Edificio 15, 90100 Palermo, Italy; 2Escola Superior de Rio Maior, CIEQV, Av. Mário Soares, 2040-413 Rio Marior, Portugal; 3Escola Superior de Desporto de Rio Maior (ESDRM-IPSantarem), Centro de Investigação em Desporto, Saúde e Desenvolvimento Humano (CIDESD), 2040-413 Rio Marior, PORTUGAL; 4Faculty of Kinesiology, University of Split, Teslina 6, 21000 Split, Croatia

**Keywords:** inclusion, equal opportunities, sport intervention, intellectual disability

## Abstract

Inactivity is a major issue that causes physical and psychological health problems, especially in people with intellectual disability (ID). This review discusses the beneficial effects of sport intervention programs (SIPs) in people with ID, and aims to provide an overview of the scientific literature in order to identify the main factors influencing the participation of people with ID in SIPs. Twelve papers were analyzed and compared. The results show a large variety in examined SIPs, concerning participants’ age and disability, intervention characteristics and context, as well as measures and findings. The main factors essential for people with ID partaking in SIPs appeared to be suitable places for the SIP development, adequate implementation of physical activity (PA) programs in school and extra-school contexts, education, and the training of teachers and instructors. The literature review highlights the relevance of using SIPs in order to improve physical and psychological health, as well as increase social inclusion in populations with ID. SIPs should be included in multifactor intervention programs. Nevertheless, the need is recognized for stakeholders to adopt specific practice and policy in promoting social inclusion in order to organize intervention strategies which are able to provide quality experiences in sport and physical activity for people with ID.

## 1. Introduction

Physical activity (PA) is a critical behavior for maintaining and improving health during one’s lifespan. A number of positive effects of PA in persons with intellectual disability (ID) include improvements in general health, such as physical fitness, bone metabolism, increased cardiovascular and respiratory muscle functions, and the control/prevention of obesity and coronary artery disease. Benefits in the social domain include functional independence and social inclusion, and benefits to psychological well-being include the increase of self-esteem, self-competence, self-efficacy, and positive self-perception [1,2,3,4].

Moreover, a large amount of research has recognized a close relationship between regular PA and brain development, particularly in the prefrontal cortical area. Several explanations for this association can be put forth: regular sport exercise programs have positive effects on the production of neurotrophins, synaptogenesis, and angiogenesis with the consequent enhancement of cognitive performance, such as the speed of information processing, working memory, planning, and behavior control strategies [5].

Following evidence from these studies, researchers and practitioners have worked to implement sport intervention programs (SIPs) aimed at obtaining the abovementioned beneficial effects for atypical populations, such as those with ID [6,7,8,9,10,11,12,13]. 

### Definition of Terms: From Physical Activity (PA) to Adapted Physical Activity (APA) and Sport Intervention Programs (SIPs)

The difference between PA and adapted physical activity (APA) is crucial to better understanding the health benefits for people with ID. PA is defined as “any bodily movement produced by skeletal muscles that requires energy expenditure” (www.who.int/dietphysicalactivity). This includes leisure, exercise, and work activities, and is based on fundamental motor skills development (walking, catching, running, throwing, etc.) [14,15]. Although the World Health Organization recommends at least 60 min of moderate to vigorous-intensity PA daily for ages ranging from 5–17 years, 150 min of moderate-intensity aerobic PA within a week for ages ranging 18–64 and above, a large number of individuals with ID are not meeting these recommendations [16]. Excluding famous examples of PA in people with disabilities (e.g., the Special Olympics (SO)), a broad gap between programs by the Special Olympics and noncompetitive sport programs has been demonstrated. There is a need to develop and implement intervention actions and educational strategies to enhance fitness levels in daily living as well as to motivate people with ID to participate in regular exercise at home and in educational settings [17].

An inactive lifestyle contributes to heightened clinical diseases and the increase of health-related complications such as higher accumulation of bone mass, type II diabetes, and functional motor impairment. Following the European Association for Research into Adapted Physical Activity, APA is a “cross-disciplinary body of practical and theoretical knowledge directed toward impairments, activity limitations, and participation restrictions in physical activity” (www.ifapa.net). The main aim is to allow and support the acceptance of individual differences and advocate enhanced access to active lifestyles and sport in people with special needs. This involves a variety of professional areas, such as physiotherapy, occupational therapy, motor rehabilitation, pediatrics, recreation, and psychology [12,15,18,19,20,21]. 

Nevertheless, sport intervention programs (SIPs) can be considered a subtype of APA; their specific characteristic is to emphasize the sport components. SIPs focus on several different sport activities, such as swimming, martial arts, cycling, movement and dance, strength, agility, and balance tasks. They are increasingly used in association with conventional methods of physiotherapy or medical interventions in people with ID [22,23,24]. The characteristics of a specific disability determine the nature of the SIP. For example, for children with autism spectrum disorder, water sports are considered a suitable exercise [25,26,27], while for individuals with Down Syndrome, the most used SIPs consist of cycling, movement and dance, strength and agility, and balance training [28]. Nevertheless, SIPs have been revealed to be especially suitable to enhance socialization by creating an inclusive environment that accommodates individual differences. Sport is a natural context that triggers social interactions and sophisticated behaviours in addition to providing proper scaffolding during cooperation [29,30,31,32,33,34,35,36,37].

However, these beneficial effects are threatened by high levels of inactivity that were found in numbers of the population with ID. Lower rates of participation in PA, in community-based sports, and in exercise programs compared to a typical development (TD) population are described in the literature. This difference is due to many reasons [38,39,40]. Often, health conditions limit their participation; physical impairment and difficulties in motor skills such as walking, running, jumping, standing, or maintaining body control are often reported [19]. Moreover, environmental barriers and lack of appropriate motivational resources impair compliance with these programs [41]. Other barriers include personal characteristics such as age, gender, motor and cognitive proficiency, and the severity of the disability. Lower intellectual abilities can reduce the ability to comprehend the exercise procedures, to plan goal-directed behaviors, and to continue exercise protocols over time. Furthermore, family overprotection towards their children, worries about clinical characteristics, as well as family socioeconomic status and availability of transportation decrease or limit the engagement of people with ID in sport programs. In summary, participation in sport programs decreases in variety, frequency, and duration with age, in coherence with higher levels of loneliness in adult age for people with ID.

Based on these theoretical considerations, this review aims to provide an overview of the scientific literature in order to identify the main factors that influence the participation in sport intervention programs (SIPs) in people with ID in the age range from 6 to 60 years. 

## 2. Materials and Methods 

The PRISMA method was followed for the review methodology and data extraction [42,43,44]. A protocol for this review was registered on PROSPERO in January 2018 (PROSPERO registration number CRD42018081672). The registration number is available at http://www.crd.york.ac.uk/PROSPERO.

### 2.1. Participants

Studies using SIPs in people with ID and chronological age between 6 and 60 years were considered. The age range from 6 to 60 years was established because the engagement in sport activities above or below these developmental phases is scarce in the population with ID. Moreover, the broad age range was a forced choice because of the paucity of studies on sport interventions among individuals with ID. In detail, 24 papers were analyzed. 

### 2.2. Procedure

To identify the literature, the following databases were searched: PsycInfo, ERIC Scopus, PubMed, Web of Science, MEDLINE, and EBSCO. The search for electronic literature databases was dated from January 1998 to February 2018. In order to ensure that no relevant studies were missed, additional studies were identified by hand-searching the reference lists of reviews and research papers. The literature search was exclusively in the English language. 

Missing papers were requested from study authors by e-mail. In each database, the following terms were searched: social inclusion, equal opportunities, ID and special needs, APA, PA, SIP. Their synonyms were identified. Duplicates and irrelevant records were eliminated. Remaining records were independently screened by two review authors to identify studies that potentially met the inclusion criteria as outlined below. 

The following inclusion criteria were adopted: studies described SIPs in people with ID;cross-sectional, cohort, experimental, and quasi-experimental;peer-reviewed studies;people with ID aged between the ages of 6 and 60.

The following exclusion criteria were adopted: studies were not published in the English language;studies were published before 1998;studies were not peer-reviewed;qualitative studies;grey literature, e.g., dissertations, conference abstracts, research reports, chapter(s) from a book, Ph.D. theses, reports on ID guidelines.

### 2.3. Data Analysis 

The initial database search identified a total of 116 papers; after a careful selection, according to the PRISMA checklist and inclusion criteria, 24 papers were analyzed for the data extraction. Figure 1 shows the flow diagram following the models of Liberati [44] and Moher et al. [45,46].

Table 1 and Table 2 present a summary of the following extracted information: Reference (Title and Country), Source (Author and Year), Study Population (Age and Sample Size), Aim of the Study, Intervention (Program), Outcome (Measures), Results and Discussion (Main issues), Quality Assessment. 

The papers’ methodological quality and the risk of bias of the reviewed studies were assessed through a 14-item quantitative checklist, as described in Table 3 [47]. The items concerned questions and methods description, outcome assessment, and conclusions. For each item, studies were scored with yes = 2, partial = 1, no = 0, or not specified (NS) when necessary. Papers were independently analyzed by two reviewers to carry out data extraction, and disagreements were discussed with a third reviewer. The results of the methodological quality evaluation were achieved as a percentage of relevant items. Fifty percent was considered as the cut-off of acceptable methodological quality (+50%) or low methodological quality (−50%).

## 3. Results

The studies included in this systematic review were found to fall in one of two research methods: randomized controlled trial design (RCTD) and non-randomized controlled trial design (N-RCTD). So, they were summarized separately (Table 1 and Table 2). In detail, 12 papers used an RCTD and 12 papers used an N-RCTD. The first ones were studies in which participants were randomly allocated into the intervention or control groups, while studies in the N-RCTD group were experimental studies without participants’ random allocation to intervention or control condition [48]. Concerning RCTD studies, eight had a methodological quality higher than 50% and four had a methodological quality lower than 50%. Regarding N-RCTD studies, nine had a methodological quality higher than 50% and three studies had a methodological quality lower than 50%.

Nevertheless, in terms of methodology, the studies covered a wide range of characteristics (e.g., participants’ age and disability, intervention characteristics, intervention context, and measures). Therefore, a narrative synthesis of results was carried out. 

In the RCTD studies that were addressed to adults, two types of measures were mainly used before and following the training period: 1. physical measures with parameters such as aerobic and anaerobic capacity, functional walking capacity, agility, muscle strength, body mass index, and motor skills; 2. well-being measures such as self-perception, self-competence, and self-esteem [38,39,49,50]. Moreover, measures of social inclusion were used: working together, creating cooperative interdependence and participation [51]. Intervention programs ranged from active lifestyle stimulation, counseling on daily PA and physical fitness training, to sport activities such as dance, swimming, and martial arts to improve participation, functioning, and health-related outcomes [52]. For children and adolescents, the parameters were physical fitness and participation associated with enjoyment [40,52]. Interventions were revealed to be quite effective in improving participation and compliance. However, in studies by Andriolo et al. [15], Slaman et al. [53], and Matthews et al. [54], the interventions were not effective in significantly increasing levels of PA. Their quality assessment is low. 

Concerning sport activities, the most used sport programs were: training to improve muscle strength, muscle endurance, physical functioning, lifestyle intervention and fitness training, aerobic exercise training, Walk Well intervention, and swimming programs. N-RCTD in adults showed positive outcomes combining physical fitness and psychosocial well-being. Physical fitness measures included walking capacity, muscular endurance and strength, flexibility, cardiorespiratory health, and functional independence. Psychosocial well-being outcomes such as exercise self-efficacy, higher life satisfaction, self-efficacy, and confidence were increased [14,55,56,57,58].

Following the aquatic program, physical fitness parameters such as cardio-respiratory endurance, strength, functional mobility, as well as skills in specific sport domains, such as aquatic proficiency [29,59,60], were used to measure the intervention effectiveness. Aquatic intervention programs were especially suitable for children [28,57,58,61]. Two studies used measures of cognitive abilities such as working memory, attention and visual–perceptual skills and executive functions [5,60,62]. 

Overall, the most used sport programs were: fitness and health education programs, physical training, and aquatic exercise program.

## 4. Discussion

This review aimed to provide an overview of the scientific literature in order to identify the main factors influencing participation in sport intervention programs (SIPs) in people with ID. Recently, there has been an increased interest in sport programs addressed to individuals with ID given the sport educational values and their efficacy in improving physical and mental health [20,36,37,42,63].

The reviewed studies showed how, for individuals with ID, sport programs positively affect the quality of life and community participation, such as satisfaction with professional services, home life, daily activities, dignity and rights, respect from others, feelings, choice, control, and family satisfaction [33,34,35]. Individuals with ID who regularly attended SIPs reported having higher opportunities for social interactions and more frequent outings into the community than sedentary peers [50,55,64]. In detail, SIPs were: training improving muscle strength, muscle endurance, physical functioning, lifestyle interventions, Walk Well intervention, swimming programs, health education programs, physical training, and aerobic and aquatic exercise programs.

Despite a wide variety of methodological variables such as participants’ age and disability, intervention characteristics, context, and measures, the main factors essential for the people with ID experiencing SIPs were revealed to concern suitable places for the SIP development, adequate implementation of PA programs in the school and extra-school contexts, and the education and training of teachers and instructors. Studies underlined the importance of providing access to physical exercise for people with ID through recreational and competitive sport and physical education curricula [20,22,36,37,42]. 

Enjoyable interventions and appealing settings are recommended to increase the repertoire of leisure skills and the level of PA in children and adolescents with ID [1,2,3,4,40,52,65]. Nevertheless, a wide range of personal factors was identified to influence the participation in PA in the lifespan. These concerned person’s disability characteristics (e.g., intellectual level, motor proficiency, social and communication competences) [49,62]. For example, the Walk Well treatment/intervention was not effective in increasing the level of PA in children with DS [54], while it was effective for other kinds of ID [49,52]. Moreover, personal factors included personal knowledge and information, interest and inclination, motivation, physical literacy skills, attitudes, and beliefs. These personal factors were influenced by interactions with socio-cultural factors at the levels of home, workplace, and community. The SIP delivery in school and extra-school contexts appeared to be essential to increase sustainable participation in PA and sport. This is because in early life and at young ages, the school is considered to be an authoritative educational agency and the privileged place of literacy to the sport. 

Moreover, inclusive education involves a person-centered, inter-cultural, and integrated educational approach that validates the experience at any age. For example, the use of group-based programs where adults with ID performed exercises together increased the program effectiveness in terms of costs and time [6,7,8,11,12,13,50,65]. 

Furthermore, the education and training of teachers and instructors are crucial to address individual needs and to plan targeted intervention programs matching the specific skills of people with ID. Expert instructors and coaches play a key role in initiating and maintaining the participation in SIPs over the time by all people given their specific education, knowledge, and awareness of strategies to favor the inclusion of people with ID.

Finally, regarding methodological aspects, 12 reviewed articles were used a RCTD and 12 used a N-RCTD. The former had the highest level of evidence because randomization allows bias to be limited and offers a rigorous instrument to investigate cause–effect links between intervention and results. However, in the population with ID, N-RCTD studies may enclose important and useful information on clinical course and prognosis, even though they provide a lower level of evidence [66]. The elevated inter-individual variability that characterizes people with ID often requires quasi-experimental or case reports studies. This was the case of the present review, showing how in many cases a great deal of preliminary educational and clinical information can be derived from studies without a control group or studies carried out on narrow groups.

## 5. Conclusions

Sport activities provide great opportunities through the medium of movement, and contribute to the overall development of individuals with ID, with the final outcome of improving and maintaining both mental and physical health. The growing emphasis on the key role of PA and sport in promoting wellbeing and social inclusion in people with ID is supported by the International Classification of Functioning, Disability and Health [67], which highlights the close relationship between body, person, and society as components characterizing human functioning, both in health and in disability conditions.

Government programs must ensure equal opportunities between people who have different disabilities and those with typical development during the life span through good practice and policy. Thus, people with ID must have the same opportunities for social inclusion as people with typical development. Indeed, SIPs have been recognized as a promising tool to promote social inclusion. High-quality sports programs offer people with ID a possibility to change lifestyle and personal perceptions in terms of wellbeing and the promotion of individual differences. From a methodological perspective, the analysis of differences between RCTD and N-RCTD studies showed the importance of implementing intervention programs with specific and structured research designs. An appropriate research design intervention—especially following a RCTD that supports evidence-based intervention programs—allows more effective results to be obtained. 

A shortcoming of this review is the large subject age range, from 6 to 60 years. This was for two reasons: 1. the engagement in sport activities above or below these ages was scarce in the population with ID; 2. there is a paucity of studies on sport interventions among individuals with ID.

To summarize, the findings of this review suggest implications on the educational and clinical fields to develop targeted evidence-based programs aiming at improving health and enhancing social inclusion through SIPs. The implementation of these intervention programs needs to account for the specificity of both disability and social/family factors to maximize the maintenance and generalization of improvements to daily living skills. In coherence with these practical implications, future research is needed to examine the the effectiveness of these programs by measuring short- and long-term improvements. Further investigations are therefore recommended to provide a richer and more complete understanding of the direction of the interrelationship between SIPs and traditional intervention methods such as physiotherapy, occupational therapy, and motor rehabilitation. Moreover, it is necessary to gather scientific evidence on the consequences of SIPs in lowering economic health costs deriving from diseases, use of medical services, and welfare assistance. 

## Figures and Tables

**Figure 1 jfmk-04-00057-f001:**
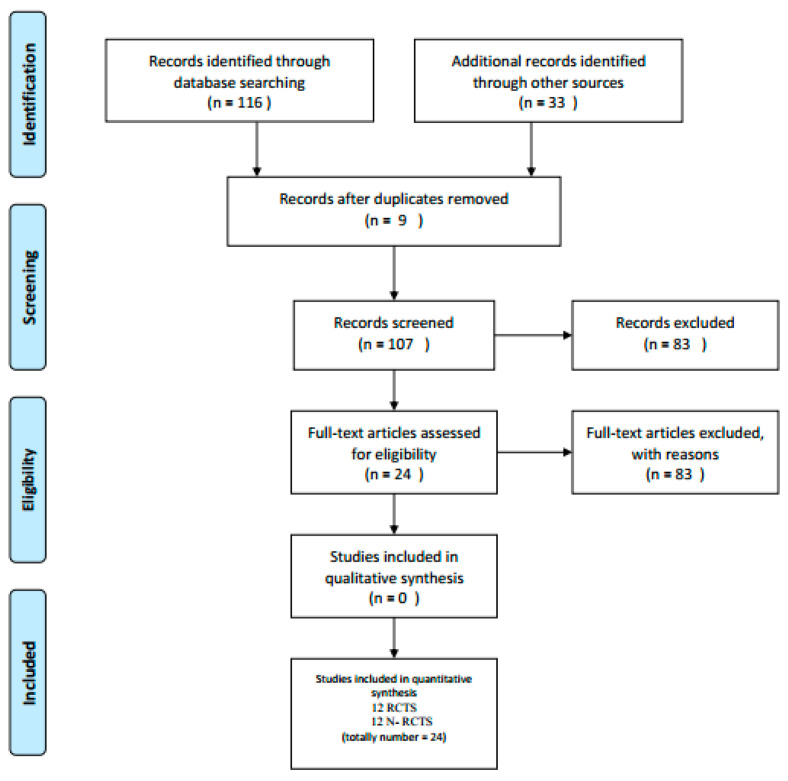
Flow diagram for the selection of papers to be included in the review.

**Table 1 jfmk-04-00057-t001:** Randomized controlled trial study (RCTS): papers analyzed in order of publication year.

Reference *Title* *Country*	Source *Author* *Year*	Study Population *Age* *Sample Size*	Aim of the Study	Intervention *Program*	Outcome *Measures*	Results and Discussion *Main Issues*	Quality Assessment
Effects of a Community-Based Progressive Resistance Training Program on Muscle Performance and Physical Function in Adults With Down syndrome: A Randomized Controlled Trial (USA)	Shields, Taylor & Dodd (2008)	Adults with Down syndrome (DS; *N* = 20) (13 men, 7 women; mean age: 26.9). Intervention group, *N* = 9; control group, *N* = 11.	To determine whether progressive resistance training improves muscle strength, muscle endurance, and physical functioning.	The intervention was organized in a supervised group progressive resistance training program with six exercises using weight machines. It was delivered twice a week for 10 weeks. All participants completed 2 to 3 sets of between 10 to 12 repetitions of each exercise until they reached fatigue.	Outcome measures were: aerobic and anaerobic capacity, agility, muscle strength, body mass index, self-perception, gross motor function, participation, and health-related quality of life (HRQOL).	The results suggest that strength training carried forward in a community setting is important for adults with DS.	63.3%
LEARN 2 MOVE 7–12 years: A randomized controlled trial on the effects of a physical activitystimulation program in children with cerebral palsy (The Netherlands)	Van Wely et al. (2010)	Children with spastic cerebral palsy (CP; *N* = 50), aged between 7 and 12 years.	To verify the efficacy of an intervention to improve physical activity (PA) by lifestyle intervention and a fitness training program (LEARN 2 MOVE 7–12).	The intervention program consisted of a 6-month physical activity and active lifestyle stimulation program associated to a 4-month fitness training program. Assessment measures were collected before the start of the intervention (T0), after the 4-month fitness training program (T4), after the 6-month lifestyle intervention (T6), and after six months of follow-up (T12).	Primary outcome of this study was physical activity. Secondary outcome measures were fitness, capacity of mobility, social participation, and HRQOL.	The results suggest that it is important for the children to maintain an active lifestyle, especially if they have better levels of fitness and to continue more physical activities in their own environment.	72.4%
Learn 2 Move 16–24: effectiveness of an intervention to stimulate physical activity and improve physical fitness of adolescents and young adults with spastic cerebral palsy; a randomized controlled trial (The Netherlands)	Slaman et al. (2010)	Adolescents and young adults with spastic CP (*N* = 60), aged between 16 and 24 years.	To verify the efficacy of an intervention to improve PA by lifestyle intervention and a fitness training program (LEARN 2 MOVE 16–24).	The intervention consisted of a 6-month program articulated in three parts: (1) counselling consisted of daily PA; (2) physical fitness training; (3) sports advice.	Outcome measures were: (1) accelerometry-based activity for measured PA level; (2) aerobic fitness; (3) neuromuscular fitness; and (4) body composition parameters determined by measuring body mass, height, waist circumference, fat mass, and lipid profile.	The results suggest that PA level can improve the fitness level of adolescents and young adults with CP by promoting a behavioral change toward a more active lifestyle.	55.6%
Promoting a healthy diet and physical activity in adults with intellectual disabilities living in community residences: Design and evaluation of a cluster-randomized intervention (Sweden)	Schäfer Elinder et al. (2010)	Adults with mild-to-moderate intellectual disability (ID; *N* = 500) aged over 18 years.	To verify the validity of a health intervention to improve diet and PA in adult age.	The intervention program was organized in three parts in 12–15 months; (1) 10 health education sessions for residents in their homes; (2) meeting with a health ambassador among the staff in each residence and the formation of a network; (3) a study circle for staff.	Outcome measures: physical activity and health measures.	The results show that that people with ID are in need of professional care, but also have the basic right to autonomy and self-determination.	46%
Aerobic exercise training programmes for improving physical and psychosocial health in adults with Down syndrome (Brazil)	Andriolo et al. (2011)	Adults with Down syndrome (*N* = 63) aged 18 years or above.	To verify the efficacy of aerobic exercise training programmed for physiological and psychosocial outcomes in adults with Down syndrome.	The intervention was an exercise training programme consisting of aerobic exercise with dynamic activities using large muscle groups. Moreover, additional instructions in health education or health awareness were given. The intervention was delivered at least three times each week for a minimum period of four weeks.	Outcome measures were: only standardized/validated scales or instruments.	No significant results were found for type of intervention (rowing vs. walking), for duration (10 vs. 16 weeks), or methods of monitoring intensity.	59%
Physical Activity Benefits of Learning to Ride a Two-Wheel Bicycle for Children With Down Syndrome: A Randomized Trial (USA)	Ulrich et al. (2011)	Children with DS (*N* = 27) aged between 8 and 15 years.	To investigate the effectiveness of a riding 2-wheel bicycle intervention program to increase PA and health-related outcomes in people with DS.	The intervention was an exercise training programme consisting of a bicycle intervention. Assessment measures were collected in the month before the intervention (pre-intervention), at 7 weeks after the intervention, and at 12 months after the pre-intervention.	Outcome measures were: leg strength and balance, anthropometric parameters (height and weight), skinfolds, and physical activity (time spent in sedentary and moderate to vigorous activity).	The results suggest that the participants who learned to ride spent significantly less time in sedentary activity at 12 months after the preintervention measurement and more time in moderate-to-vigorous physical activity than participants in the control group.	61.1%
Walk well: a randomised controlled trial of a walking intervention for adults with intellectual disabilities: study protocol (Scotland)	Mitchell et al. (2013)	Adults with ID (*N* = 40), aged over 18 years.	To evaluate the effectiveness of walking interventions (Walk Well intervention) impact on PA levels, health, and wellbeing in adults and older adults with ID.	The intervention consisted of three PA consultations and an individualized 12-week walking programme. Assessment measures were collected at baseline, post intervention (three months from baseline), and at follow-up (three months post intervention and six months from baseline).	The outcome measures were physical activities such as dance, swimming, and martial arts, and their improvement of participation, functioning, and health-related outcomes.	A significant increase was found in walking for participants who undertook the PA consultations and individualised walking programme.	67%
Can a lifestyle intervention programme improve physical behaviour among adolescents and young adults with spastic cerebral palsy? A randomized controlled trial (The Netherlands)	Slaman et al. (2014)	Adults with spastic CP (*N* = 57), aged between 16 and 25 years.	To evaluate the effectiveness of a lifestyle intervention programme on physical behavior in persons with CP.	The intervention consisted of a 6-month lifestyle programme consisting of fitness training and counselling on physical behaviour and sports participation.	Outcome measures of physical behaviours were objectively measured using ambulatory activity monitors. Self-reported PA was measured using the Physical Activity Scale for Individuals with Physical Disabilities.	The results suggest that the intervention did not affect the objectively measured PA during the intervention or at follow-up in adolescents and young adults with spastic CP.	48.5%
Changing Attitudes Toward Disabilities Through Unified Sports (USA)	Sullivan et al. (2014)	Adults with ID (*N* = 16), aged between 17 and 21 years.	To examine the efficacy of a unified swimming program to: 1. result in more positive attitudes of persons without disabilities towards persons with ID; 2. Positively affect persons without disabilities; 3. Positively affect persons with ID.	The intervention consisted of four sessions over a 6-week program period involving swimming activities.	Outcome measures were related to the shared objective of arriving at a common outcome serving as a powerful force to motivate working together, creating cooperative interdependence, and facilitating positive attitude change.	The results suggest that on a revision attitudes inventory, the intervention group participants displayed significant increases in positive attitudes from pre- to post-test, whereas the control group participants did not.	47.4%
Embedding sustainable physical activities into the everyday lives of adults with intellectual disabilities: A randomised controlled trial (Australia)	Lante et al. (2014)	Adults with ID (*N* = 90) aged between 18 and 55. Three groups: (1) a lifestyle physical activity group (*N* = 30); (2) a structured exercise group (*N* = 30); (3) a usual care control group (*N* = 30).	To compare two approaches to increase short-term (3-month) and long-term (9-month) PA outcomes in adults with ID: a lifestyle physical activity (light–moderate intensity, LMI) approach versus a structured exercise (moderate–vigorous intensity, MVI) approach.	The intervention (LMI—Lifestyle intervention physical activity or a structured MVI—Exercise group) consisted of a 12-week programme delivered by exercise specialists in the community with disability service staff, after which intervention continued for 6 months, delivered by disability service staff only.	Primary outcomes were aerobic fitness, 12-h energy expenditure, and proxy-reported everyday physical activity. Secondary outcomes were objectively assessed physical activity and sedentary behaviour, intervention compliance, functional walking capacity, participation in domestic activities, muscle strength, body composition, psychosocial outcomes, quality of life, and health care costs.	The results suggest the effectiveness and sustainability of the two approaches in increasing physical activity and exercise among adults with ID.	56%
Effectiveness of a walking programme to support adults with intellectual disabilities to increase physical activity: Walk Well cluster-randomised controlled trial (Scotland)	Melville et al. (2015)	Adults with ID (*N* = 102) aged over 18 years.	To verify the efficacy of a behavior change programme to increase walking and reduce sedentary behaviors in adults with ID.	Walk Well was a 12-week intervention programme consisting of three face-to-face PA consultations based on behaviour change techniques, written resources for participants and carers, and an individualised, structured walking programme.	The primary outcome measure was the mean step count per day between baseline and 12 weeks, evaluated with accelerometers. Secondary outcome measures were percentage time per day spent sedentary and in moderate–vigorous physical activity (MVPA), body mass index (BMI), and subjective well being.	The results suggest that this is the first published trial of a walking program for adults with intellectual disabilities with a positive effect.	51%
Process evaluation of the Walk Well study: a cluster-randomised controlled trial of a community based walking programme for adults with intellectual disabilities (Scotland)	Matthews et al. (2016)	Adults with ID (*N* = 102) aged over 18 years.	To evaluation the efficacy of a community-based walking intervention for adults with ID.	The Walk Well intervention of 12-week PA was a single-blind cluster-randomised controlled trial PA consultation-led walking intervention. The intervention group was delivered three PA consultations with a walking advisor at baseline, 6, and 12 weeks.	The outcome measures were change in daily step count at 12 weeks. Process evaluation used both qualitative interviews with stakeholders, quantifiable data collected during the intervention, and a qualitative interview with participants.	The results show that Walk Well was not effective in significantly increasing levels of physical activity.	44%
Total:	12	papers					

**Table 2 jfmk-04-00057-t002:** Non-randomized controlled trial study (N-RCTS) papers analyzed in order of publication year.

Reference *Title and Country*	Source *Author* *Year*	Study Population *Age* *Sample Size*	Aim of the Study	Intervention *Program*	Outcome *Measures*	Results and Discussion *Main Issues*	Quality Assessment
Attitudinal and Psychosocial Outcomes of a Fitness and Health Education Program on Adults With Down Syndrome (USA)	Heller et al. (2004)	Adults with Down syndrome (*N* = 53) aged 30 years and older.	To evaluate the effectiveness of a fitness and health education program to improve attitudinal and psychosocial outcomes in adults with Down syndrome.	The intervention program consisted of a 12-week, 3 days per week health promotion program lasting 2 h a day. In particular, a 1-h exercise class and 1-h health education component.	The outcome measures were attitudes towards exercise (cognitive–emotional barriers, outcomes expectations, and performance self-efficacy), and psychosocial well-being (community integration, depression, and life satisfaction).	The results suggest improved attitudes towards exercise, increased exercise self-efficacy, more positive expected outcomes, fewer cognitive–emotional barriers, higher life satisfaction, and lower depression.	80.3%
Can physical training have an effect on well-being in adults with mild intellectual disability? (Israel)	Carmeli et al. (2005)	Adults with ID (*N* = 22) aged between 54 and 66 years.	To investigate the effectiveness of physical training in adult people with ID, particularly strength and physical/psychological well-being.	The intervention was focused on the improvement of muscle strength, and general exercises. The training program for the two groups was performed 3 days a week during six consecutive months. The physical training program took place three times a week for six months.	The outcome measures were: self-efficacy, confidence, life satisfaction.	The results suggest positive effects of physical activity on mental health and psychological well-being for adults aged 60 years and older. Moreover, the results suggest the important role of physical training to enhance locomotor performance and perception of well-being among “older” adults with ID.	66%
Physical activity and its determinants among adolescents with intellectual disabilities (Taiwan)	Lina et al. (2009)	Adolescents with ID (*N* = 351) aged between 16 and 18 years.	To describe the regular PA prevalence and examine its determinants among adolescents with ID in Taiwan.	The intervention program involved the people with ID that had regular physical activity habits. The physical activities included were walking, sports, and jogging at least exercise three times per week, 30 min per time.	The outcome measures were the physical education class in school, involving outdoor activities such as walking, jogging, sports, cycling, swimming, and dancing.	The results suggest the need to sustain sport motivation to participate in PA for people with ID in order to increase the positive effects of PA on this population.	42.1%
Summative evaluation of a pilot aquatic exercise program for children with disabilities (USA)	Fragala-Pinkham et al. (2010)	Child participants with a variety of diagnoses (*N* = 16): autism spectrum disorders (ASDs; *N* = 6), cerebral palsyespastic diplegia (*N* = 1) and hemiplegia (*N* = 1), Down syndrome (*N* = 2), myelomeningocele with the lesion at the lumbar spine level (L4–5) (*N* = 1) and the sacral level (S1–2) (*N* = 1), developmental delay (*N* = 2), nonverbal learning disorder (*N* = 1), and oto-palatal-digital syndrome (*N* = 1).	To evaluate the efficacy of a pilot aquatic exercise program in children with disabilities.	The aquatic exercise consisted of a program warm-up (3–5 min), aerobic conditioning (20–30 min), strengthening exercises (5–10 min), and cool-down (3–5 min), 2 times per week for 14 weeks. The pool sessions lasted about 45 min.	The outcome measures were cardiorespiratory endurance, strength, functional mobility, and safety.	The results suggest that the children improved their swimming skills by increasing their PA levels during the program and maintaining this six months after the program ended.	56.3%
The efficacy of an aquatic program on physical fitness and aquatic skills in children with and without autism spectrum disorders (Taiwan)	Pan (2011)	Children with ASD (*N* = 15) and their siblings (*N* = 15) aged between 7 and 12 years.	To evaluate, in children with ASD and their siblings without a disability, the effectiveness of an aquatic program on physical fitness and aquatic skills.	The intervention program was organized in 32 weeks, with 14 weeks aquatic program, 14 weeks control, and 4 weeks assessment and transition. In the first phase over 14 weeks, 14 children (group A: ASD, *n* = 7; siblings, *n* = 7) received the aquatic program, while 16 children (group B: ASD, *n* = 8; siblings, *n* = 8) did not. The second phase lasted 14 weeks.	Outcome measures were aquatic skills and physical fitness components.	The results suggest that aquatic skills and physical fitness increased following the aquatic program. Moreover, the results highlight the need to develop and implement intervention programs to enhance motor skills and physical fitness components for children with ASD and their siblings.	47.1%
Effects of an exercise programme on anxiety in adults with intellectual disabilities (Italy)	Carraro et al. (2012)	Adults with mild ID (*N* = 18) and moderate ID (*N* = 9) aged between 31 and 49 years.	To evaluate the efficacy of a short-term exercise programme on anxiety states in adults with ID.	The intervention program was organized into two 1-h sessions per week over 12 consecutive weeks. There were three phases: an initial warm-up, a central phase in which the main topic was developed, and a cool-down including group discussion and individual comments circle time technique.	The outcomes measures were flexibility, muscular endurance and strength, cardiorespiratory health and functional independence, and functional and musculoskeletal health, such as walking capacity.	The results suggest the crucial role of exercise in people with ID as a strategy to promote mental health and wellbeing.	77%
The effect of peer- and sibling-assisted aquatic program on interaction behaviors and aquatic skills of children with autism spectrum disorders and their peers/siblings (Taiwan)	Chu et al. (2012)	Children with ASD (*N* = 21) aged between 7 and 12 years old.	To evaluate the efficacy of an aquatic program in children with autism spectrum disorders (ASDs) to increase peer- and sibling-assisted learning on interaction behaviours and aquatic skills.	The intervention consisted of different times for each session. All participated in 16-week aquatic settings under three instructional conditions (teacher-directed, peer/sibling-assisted, and voluntary support).	The outcome measures were interaction behaviours and aquatic skills.	The results suggest that all children with ASD and their typical development (TD) peers/siblings significantly increased their aquatic skills after the program. The use of TD peer/sibling-assisted learning is a convenient instructional strategy.	54.2%
Improvement of gross motor and cognitive abilities by an exercise training program: Three case reports (Italy)	Alesi et al. (2014)	Children with Down syndrome (*N* = 3) with chroological age of 10.3, 14.6, and 14 years, respectively and a mental age of less than 4 years.	To evaluate the effectiveness of an integrated exercise training program to improve motor and cognitive abilities in three children with Down syndrome.	The integrated exercise training program was distributed twice a week for a period of 2 months. This consisted of the following stages: a social interaction phase between child, coach, and parents (about 5 min) to enhance the motivation to participate; a warm-up period (~5 min); a central training period (~40 min) including two nursery rhyme games and several activities aimed at improving basic motor abilities such as running, jumping, throwing, and rolling; a cool-down period (~5 min); and a feedback phase (~5 min) to explore the child’s satisfaction level.	The outcome measures were chronological and mental age, body weight, height, gross motor skills, working memory, and attention skills.	The results suggest that in individuals with DS there is an important relation between motor and cognitive domains. For children and parents it is important to plan intervention programs based on the simultaneous involvement in order to promote an active lifestyle.	77.2%
Effects of declared levels of physical activity on quality of life of individuals with intellectual disabilities (USA)	Blick et al. (2015)	Participants with ID (*N* = 788) aged between 11 and 92 years.	To evaluate the impact of physical fitness on quality of life by comparing sedentary individuals with ID and individuals with ID who reported frequently exercising, as more than 12 times per month.	The intervention consisted of physical activity for individuals with physical disabilities. The participants were organized in three groups based on frequency of exercise participation: never exercises (*n* = 335, 42.5%), exercises 1–11 times per month (*n* = 237, 30.1%), and exercises more than 11 times per month (*n* = 216, 27.4%).	The outcome measures were employment, the quality of interpersonal relationships and inclusion in community activities, satisfaction with services received, and choice and control over daily activities.	The results suggest that the physical fitness enhanced the health and psychosocial well-being of individuals with ID who maintained a physically active lifestyle compared to individuals who did not report exercising. Individuals with a regular exercise habit had more frequent outings into the community than did their sedentary peers.	47%
Teaching advance movement exploration skills in water to children with autism spectrum disorders (Turkey)	Yanardag et al. (2015)	Children with ASD (*N* = 3) with chronological age of 6 years.	To evaluate the effectiveness of the “most to least” Prompting (MLP) program to improve the learning processes of advance movement exploration skills in water for children with ASD.	The intervention consisted of MLP exercises to teach three different aquatic skills, essential for movement exploration in water and swimming, in a one-to-one training format over three sessions per week.	The outcome measures were physical activity’s stimulation of the improvement of physical fitness, motor performance, self-esteem, behaviour, and social outcomes.	The results suggest MLP was effective in teaching advance movement exploration skills in water to children with ASD. It is an important enjoyable intervention and an appealing setting to improve leisure skills and PA level for children with ASD.	64%
The effectiveness of racket-sport intervention on visual perception and executive functions in children with mild intellectual disabilities and borderline intellectual functioning (Taiwan)	Chen et al. (2015)	Children with ID (*N* = 91) aged 6–12 years.	To evaluate the effects of table tennis training (TTT) versus standard occupational therapy (SOT) on visual perception and executive functions in school-age children with mild ID and borderline intellectual functioning (BIF).	The SOT and TTT programs were administered 60 min per session, three times a week, for 16 weeks.	The outcome measures were racket sports. In particular, the correlation between improved visual–perceptual and executive functions and school function.	The results suggest contributing to the treatment of cognitive/perceptual problems in children with mild ID and BIF with table tennis, which is a good alternative therapy.	71.2%
Design and methods of a multi-component physical activity program for adults with intellectual disabilities living in group homes (Taiwan)	Chow et al. (2016)	Adults with ID (*N* = 62) aged between 18 and 55 years.	To evaluate the efficacy of intervention program PA levels in adults with ID living in group homes through multi-component physical activity (PA).	Thirty exercise sessions were run in groups over a 10-week period and three educational lessons were given to increase self-efficacy and social support for PA. In addition, staff training in exercise and advice concerning PA policies were provided to the caregivers working in the group homes.	The outcome measures were physical fitness variables, physical activity measures, and psychosocial variables. In particular, measurements were taken at baseline (week 0), post-intervention (week 11), and at the 10-week follow-up.	The results suggest that in adults with ID, it is important to promote active lifestyles.	64%
Total:	12	papers.					

**Table 3 jfmk-04-00057-t003:** Quality assessment checklist.

Criteria	Yes (2)	Partial (1)	No (0)	NS
1 Question/objective sufficiently described?		X		
2 Study design evident and appropriate?	X			
3 Method of subject/comparison group selection or source of information/input variables described and appropriate?		X		
4 Subject (and comparison group, if applicable) characteristics sufficiently described?		X		
5 If interventional and random allocation was possible, was it described?	X			
6 If interventional and blinding of investigators was possible, was it reported?				X
7 If interventional and blinding of subjects was possible, was it reported?				X
8 Outcome and (if applicable) exposure measure(s) well-defined and robust to measurement/misclassification bias? Means of assessment reported?	X			
9 Sample size appropriate?		X		
10 Analytic methods described/justified and appropriate?			X	
11 Some estimate of variance is reported for the main results?			X	
12 Controlled for confounding?		X		
13 Results reported in sufficient detail?		X		
14 Conclusions supported by the results?		X		

Adapted from Kmet et al. [47]. Legend: NS not specified.

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
