# Peer review of "Sport Intervention Programs (SIPs) to Improve Health and Social Inclusion in People with Intellectual Disabilities: A Systematic Review"

_jfmk, 2019, doi:10.3390/jfmk4030057_

Round 1

Reviewer 1 Report

The work provides a coherent overview of the previously published results on the importance of physical activity to improve health and social inclusion in people with intellectual disabilities (Sport Intervention Programes - SIP). The detailed analysis includes 24 links, presented through tables, and some of them are referred to in the discussion. Overall, 61 sources were used in this survey, of which only a minor part (15) originates in the period older than 10 years (up to and including year 2009).

The analysis provided in the text is presented adequately, eliciting no doubts regarding correct and proper presentation of the results. Nevertheless, in terms of methodology, the criteria covered in the study are rather wide and extensive:

o   age range (7-60 yrs)

o   type of disability - Down syndrome, cerebral palsy, a general, resp. non-specified range of intellectual disabilities, autism spectrum disorders

o   sample size ranging from n=3 (Alesi et al. (2014)) to n=788 (Blick et al. (2015), respectively, size not indicated (Andriolo et al. (2011), Yanardag et al. (2015))

o   intervention program - duration and content of the program

o    various outcome measures

The study would gain higher quality if the methodology were set more strictly (i.e. a narrower range of the above variables)

Recommendations:

o    in methodology: specify more closely the reason for excluding particular studies (with reference to Figure 1: Flow Diagram

o    supplement/provide outcome measures in studies by Schäfer Elinder et al. (2010), Ulrich et al. (2011) and Pan (2011)

o   specify sample size in Andriolo et al. (2011) and Yanardag et al. (2015)

o   in the review: Shields, Taylor & Dodd, (2008); Schäfer Elinder et al. (2010); Ulrich et al. (2011), add to the list of literature the authors whose articles have been used

o   make corrections to authors listed in References, resp. in quotations (incorrect first author's name and year): Slaman et al. (2014) is listed in References under no. 51 as Slamn et al. (2015) and the author Lina et al. (2009) is listed in References under no. 16 as Lin et al. (2010)

Author Response

Recommendations:

          in methodology: specify more closely the reason for excluding particular studies (with reference               to Figure 1: Flow Diagram

We included a detailed section Inclusion and Exclusion criteria (lines 135-141).

·         supplement/provide outcome measures in studies by Schäfer Elinder et al. (2010), Ulrich et al. (2011) and Pan (2011)

We have expanded and detailed the outcome measures in the two studies required

·         specify sample size in Andriolo et al. (2011) and Yanardag et al. (2015)

We specified the sample size in the studios of: Andriolo et al. (2011) and Yanardag et al. (2015)

·         in the review: Shields, Taylor & Dodd, (2008); Schäfer Elinder et al. (2010); Ulrich et al. (2011), add to the list of literature the authors whose articles have been used

We have added to the end of the list of literature to the list of literature the authors : lines 470- : Shields, Taylor & Dodd, (2008); 474- Schäfer Elinder et al. (2010); 478- Ulrich et al. (2011).

·         make corrections to authors listed in References, resp. in quotations (incorrect first author's name and year): Slaman et al. (2014) is listed in References under no. 51 as Slamn et al. (2015) and the author Lina et al. (2009) is listed in References under no. 16 as Lin et al. (2010)

We have made corrections to authors listed in References. In particular, lines 443 - Slamn et al. (2014); 340- Lina et al. (2009).

Reviewer 2 Report

Thank you for the opportunity to review your manuscript. While it addresses an important topic, the writing is difficult to read in places, and the paper does not flow well.

Below are some comments to address to improve the writing:

General comments: Need to proofread and check for grammar throughout. The discussion section is short, and some of the results section sounds more like the discussion.

Spell out ID in the abstract. Also, you spell out SIP twice. You can remove the second one.

Intro: Line 39  - PA is more of a behavior than a tool – consider rephrasing it as such. Perhaps “PA is a critical behavior for maintaining or improving health…”?

Line 45: This line is double-indented. You can remove the second indent. Also, what is “large research”? Does this mean large amounts of research, or a large research study? Please clarify.

Line 46: What is “sport beneficial effects”? Do you mean the beneficial effects of physical activity on neurotrophins, etc? Please clarify and rephrase

Lines 50-52: These sentences need a better connection to the previous paragraphs. Do you mean as a consequence of the research showing positive benefits of activity on cognitive outcomes, there are numerous SIP implemented for atypical populations, such as those with ID? If so, phrase it that way.

Line 57: This is not the standard definition of physical activity generally accepted in the literature. Are you using a definition of PA as defined for special populations?

Lines 60-67: You need to define APA, not just describe it

Lines 68-72: Need a connection between SIP and the preceding paragraph. How are SIP and APA related?

Line 73: This sentence is incomplete. Implemented where, or why or how?

Lines 73-81: You can probably just describe one example, rather than three.

Lines 104-195: use the past tense for the methods (were instead of are, was instead of is)

Lines 197-201: These lines should go in the introduction, not the results

Lines 207-211: Please re-word. This is a long run-on sentence that is confusing.

Lines 211-213: This sounds like it should be placed in the discussion, not results.

Table 3: What does NS mean in the 4th column?

Author Response

General comments: Need to proofread and check for grammar throughout. The discussion section is short, and some of the results section sounds more like the discussion.

A native English speaker read and amended the revised version.

The discussion was enlarged and part of results was moved to Discussion. A section with Conclusions was added.

Spell out ID in the abstract. Also, you spell out SIP twice. You can remove the second one.

ID was spelled one and the second SIP spelling was removed

Intro: Line 39  - PA is more of a behavior than a tool – consider rephrasing it as such. Perhaps “PA is a critical behavior for maintaining or improving health…”?

Line 45: This line is double-indented. You can remove the second indent. Also, what is “large research”? Does this mean large amounts of research, or a large research study? Please clarify.

Line 46: What is “sport beneficial effects”? Do you mean the beneficial effects of physical activity on neurotrophins, etc? Please clarify and rephrase

Lines 50-52: These sentences need a better connection to the previous paragraphs. Do you mean as a consequence of the research showing positive benefits of activity on cognitive outcomes, there are numerous SIP implemented for atypical populations, such as those with ID? If so, phrase it that way.

Line 57: This is not the standard definition of physical activity generally accepted in the literature. Are you using a definition of PA as defined for special populations?

Lines 60-67: You need to define APA, not just describe it

Lines 68-72: Need a connection between SIP and the preceding paragraph. How are SIP and APA related?

Line 73: This sentence is incomplete. Implemented where, or why or how?

Lines 73-81: You can probably just describe one example, rather than three.

Lines 104-195: use the past tense for the methods (were instead of are, was instead of is)

Lines 197-201: These lines should go in the introduction, not the results

Lines 207-211: Please re-word. This is a long run-on sentence that is confusing.

Lines 211-213: This sounds like it should be placed in the discussion, not results.

All suggested changes from line 39 to line 213 were done

Table 3: What does NS mean in the 4th column?

NS means not specified. We added a legenda in table 3

Reviewer 3 Report

A review study should include more elaborative discussion (using the attached figure and tables) and better summarize the conclusion. After reading the introduction and examining the attachments, I have a feeling of poor discussion as it didn't meet my expectation. Therefore, I would strongly recommend you to develop both parts of the manuscript.   

Author Response

A review study should include more elaborative discussion (using the attached figure and tables) and better summarize the conclusion. After reading the introduction and examining the attachments, I have a feeling of poor discussion as it didn't meet my expectation. Therefore, I would strongly recommend you to develop both parts of the manuscript.   

 The discussion was enlarged and part of results was moved to Discussion. A section with Conclusions was added.

Round 2

Reviewer 2 Report

Thank you for the opportunity to review the revised version of your manuscript. While improved, there are additional edits that could be made to further enhance the suitability of this work for publication.

General comment: Throughout the entire paper, please have the grammar proofread and edited for use of proper English. There are many minor errors in verbiage, tense, and general grammatical issues that make this paper difficult to read. 

Specific comments:

Line 70 - Can you provide a citation for the large numbers who do not meet the recommendation? Can you make a more convincing case that people with ID are not very active? There are numerous programs that exist (Special Olympics, etc.) to increase their sport participation.

Lines 80-81: Not sure what this means.

Line 107 - It feels like a sub-aim is examining differences by RCT and NRCT. If so, please state that.

Line 185: What is "methodological quality?" Please define.

Discussion Please provide a summary/discussion of the RCT and NRCT differences, which seem like an important sub-aim of this study - Do the authors have recommendations for which is a better or more effective study design?

Line 273 - what are "typical people"? Be careful mentioning subjective terms like that.

Conclusions: Could you provide directions for future research?

Author Response

Response to Reviewer 2 Comments

General comment: Throughout the entire paper, please have the grammar proofread and edited for use of proper English. There are many minor errors in verbiage, tense, and general grammatical issues that make this paper difficult to read. 

The paper has been entirely revised as concern grammar and other linguistic aspects

Specific comments:

Line 70 - Can you provide a citation for the large numbers who do not meet the recommendation? Can you make a more convincing case that people with ID are not very active? There are numerous programs that exist (Special Olympics, etc.) to increase their sport participation.

Lines 71-76 Difference between Special Olympics and daily fitness levels were introduced and supported by new references.

Lines 80-81: Not sure what this means.

Lines 86-87. The difference between APA and SIP was re-written

Line 107 - It feels like a sub-aim is examining differences by RCT and NRCT. If so, please state that.

Many thanks for this observation. This is not a sub-aim. It was moved to results section (Lines 177-184).

Line 185: What is "methodological quality?" Please define.

Lines 164-169. A more extensive description of methodological quality was added

Discussion Please provide a summary/discussion of the RCT and NRCT differences, which seem like an important sub-aim of this study - Do the authors have recommendations for which is a better or more effective study design?

Lines 261-269. The difference between RCT and N-RCT was discussed.

Line 273 - what are "typical people"? Be careful mentioning subjective terms like that.

Lines 281-282 We changed in “people with typical development”

Conclusions: Could you provide directions for future research?

Lines 296-303: future research directions were added
